# Nasal Lymphoma with Low Mitotic Index in Three Cats Treated with Chlorambucil and Prednisolone

**DOI:** 10.3390/vetsci9090472

**Published:** 2022-09-01

**Authors:** Karen W. L. Ng, Julia A. Beatty, May P. Y. Tse, Antonio Giuliano

**Affiliations:** 1CityU Veterinary Medical Centre, City University of Hong Kong, Hong Kong; 2Department of Veterinary Clinical Sciences, Jockey Club College of Veterinary Medicine, City University of Hong Kong, Hong Kong; 3Centre for Animal Health and Welfare, City University of Hong Kong, Hong Kong; 4VDL Veterinary Diagnostic Laboratory, City University of Hong Kong, Hong Kong

**Keywords:** cats, chlorambucil, chemotherapy, grade, feline, nasal lymphoma

## Abstract

**Simple summary:**

Nasal lymphoma is the most common type of cancer of the nasal cavity in cats. Nasal lymphoma in cats is usually treated with injectable chemotherapy and/or radiation therapy if it is localized to the nasal cavity. These treatments require significant commitments in both time and finances from the owner. In this case series, we report three cats with nasal lymphoma with a low mitotic index, treated with oral chlorambucil and prednisolone, with two of them achieving a relatively good survival time.

**Abstract:**

Lymphoma is the most common tumor of the nasal cavity in cats. Commonly used treatment modalities are radiotherapy and chemotherapy, or both. Typical chemotherapy protocols used in cats with nasal lymphoma are COP (cyclophosphamide, vincristine prednisolone) and CHOP (cyclophosphamide, doxorubicin, vincristine and prednisolone). Thus far, the use of single-agent chlorambucil in nasal lymphoma has been evaluated in a single case report. We report a case series of three cats with B cell nasal lymphoma, two cats with intermediate cell, and one large cell, all with a low mitotic index (MI) of less than 5 mitotic figures per ×400 field, treated with chlorambucil and prednisolone. Two of the cases achieved a long disease-free interval, while the one with the highest MI did not. Protocols using chlorambucil and prednisolone may have potential as a first-line therapy for feline nasal lymphoma cases with a very low mitotic index.

## 1. Introduction

Lymphoma is the most common tumor of the nasal cavity in cats [1,2,3]. Clinical signs include chronic nasal discharge, which may be blood-tinged, sneezing and stertor. In advanced cases, facial deformity, exophthalmos and local or distant extension beyond the nasal cavity may occur [4,5].

Histologically, feline nasal lymphomas are usually high-grade, with 60–70% being of B cell origin [3,5,6]. Low-grade/small-cell nasal lymphoma is rarely reported. Grade is not considered an important prognostic factor in nasal lymphoma [7], with low-/intermediate- and high-grade lymphoma often treated in the same way [4]. Nasal lymphoma in cats is typically treated with COP (cyclophosphamide, vincristine and prednisolone) or CHOP (cyclophosphamide, doxorubicin, vincristine and prednisolone) protocols. Localized lymphoma of the nasal cavity can also be treated successfully with radiotherapy. However, around 20–30% of patients develop extra-nasal involvement, such as metastasis to the regional lymph nodes and kidneys [5,8].

The survival time of cats with nasal lymphoma treated with COP, CHOP chemotherapy protocols or radiotherapy is variable, and is reported to be between 116 and 750 days depending on the study [4,9,10].

Mild to moderate side effects are common with COP or CHOP protocols. dos Santos Cunha et al. reported around 83% of patients developing mild side effects, and a small percentage of patients (2%) can develop severe life-threatening complications [11].

Radiation therapy, COP and CHOP chemotherapy require frequent visits to the clinic and significant owner commitment. Side effects are common and may require dose reduction and/or supportive treatments or cessation of treatment, but are rarely life-threatening. The benefit of radiotherapy versus chemotherapy is not proven and, at least in one study, there was no significant difference in survival times in cats with nasal lymphoma treated with radiotherapy, chemotherapy or both [4].

Chlorambucil, an oral nitrogen mustard alkylating agent, combined with prednisolone treatment, has mainly been described to treat feline low-grade lymphoma and, most commonly, for (small-cell or low-grade) gastrointestinal lymphoma [12,13].

The potential benefits of using chlorambucil and prednisolone to treat low-grade nasal lymphoma include less side effects, lower costs and less intensive treatment and monitoring schedules, but its use in cats has not been fully investigated. Single-agent chlorambucil has been reported in only one case report in a cat with large B cell nasopharyngeal lymphoma with a low mitotic index (MI) of 5 in 10 per ×400 fields [14].

In this case series, we describe the use of chlorambucil and prednisolone in three feline intermediate-to-large B cell, low-grade nasal lymphomas.

## 2. Materials and Methods

Cases of feline nasal lymphoma treated with chlorambucil and prednisolone were retrospectively reviewed. The histopathology samples were reviewed by a certified pathologist and graded based on the classification of canine malignant lymphomas according to the World Health Organization criteria by Valli et al., which consider low-grade lymphoma as tumors with a low MI of less than 5 per ×400 field [15,16,17]. This classification has been used in other studies, mainly for grading cats with intestinal and upper respiratory lymphoma [7,18].

### 2.1. Description of Cases

#### 2.1.1. Case N.1

Aged 8 years and 11 months, a female neutered domestic short-haired cat was presented for progressive stertor, sneezing and chronic nasal discharge of a three-month duration. On presentation, the cat was bright, alert and responsive, but markedly underweight at 1.48 kg, with a body condition score (BCS) of 2/9. On physical examination, the cat had bilateral mucous purulent nasal discharge and nasal airflow was reduced bilaterally. No facial deformity, exophthalmos or mass effect on retropulsion of both eyes were present. Regional and all peripheral lymph nodes were within normal limits. Thoracic auscultation, abdominal palpation and the reminder of the physical examination were unremarkable. The cat had a history of well-controlled hyperthyroidism on carbimazole and antibiotic (metronidazole)-responsive diarrhea. She was feline leukemia virus (FeLV)-negative, but feline immunodeficiency virus (FIV)-positive on the IDEXX^®^ snap FIV/FELV combo test.

Hematology and biochemistry were unremarkable, and serum total T4 was within normal limits. Computed tomography (CT) of the head and neck revealed an occlusive mass from the choanae and nasopharyngeal meatus to the caudal margin of the hard palate, bulging into the rostral portion of the nasopharynx. There was no evidence of osteolysis or abnormal periosteal reaction. There was moderate associated rhinitis and sphenoidal sinusitis and mild mandibular and superficial cervical lymphadenopathy.

Posterior rhinoscopy revealed a multi-lobulated bilateral intranasal mass (2.3 cm × 1.9 cm × 0.7 cm). Histopathology revealed diffuse aggregate of sheets of neoplastic cells supported by a light fibrovascular stroma. The cells were round to elongate with minimal cytoplasm and vesicular pleomorphic nuclei and 1 to 2 distinct, magenta, medium nucleoli. The nuclei of the neoplastic cells were 2 to 4 times the size of a red blood cell. The mitotic rate was 3–4 per ×400 field. Hematoxylin and eosin (H&E)-stained sections suggested lymphoma (Figure 1 and Figure 2) and immunohistochemistry staining was positive for CD79a (Biocare Medical, Pacheco, CA, USA, PM067AA) (Figure 3) and negative for CD3 stain (Dako, Carpinteria, CA, USA, A0452), confirming B cell origin. A diagnosis of nasal lymphoma, diffuse, large-cell type, B cell, was made. A second pathologist reviewed the histopathology slides and confirmed the diagnosis. Low grade was concluded due to the relatively low MI based on Valli’s classification of lymphoma in dogs [15,16,17].

The cat markedly improved after starting prednisolone treatment (10 days), with no significant stertor or nasal discharge. The owners declined COP/CHOP chemotherapy treatment and the cat was treated with oral chlorambucil 2 mg three times a week (5–6 mg/m^2^ per day) and prednisolone 5 mg once daily. Due to clinical improvement after 7 weeks, chlorambucil was later reduced to 2 mg twice a week and prednisolone was reduced to 2.5 mg once daily. Rechecks were carried out monthly and no clinical side effects or hematological and biochemistry abnormalities related to chlorambucil were noticed.

Progression based on worsening of her clinical signs occurred after three months. The owner again declined injectable chemotherapy and elected to continue only with another oral chemotherapy option. The cat was started on an oral LPP protocol (lomustine, procarbazine and prednisolone) for another 3 months. Unfortunately, the patient progressed again 3 months later. The owners finally agreed to try injectable chemotherapy (CHOP). The nasal lymphoma again progressed 3 months later. In addition to the local progression of the lymphoma, the patient developed severe pancytopenia, suspected to be lymphoma-related. However, other causes could not be ruled out as the owner declined any further investigations, and the cat was eventually euthanized. This patient had a chlorambucil and prednisolone disease-free interval (DFI) of 3 months and a total survival time of 9.5 months.

#### 2.1.2. Case N.2

Aged 7 years and 11 months, a male neutered domestic short-haired cat was presented with a six-month history of intermittent sneezing, nasal discharge and occasional epistaxis. The cat was previously treated with antiviral famciclovir, with no significant response. On presentation, the cat was bright, alert and responsive, and overweight at 5.74 kg, with a BCS of 7/9. On physical examination, the cat had mild bilateral serous nasal discharge, epiphora and reduced air flow from the left nasal passage; no facial deformity and no exophthalmos or mass effect on retropulsion of the eyes were present. Regional and all peripheral lymph nodes were within normal limits; abdominal palpation, thoracic auscultation and the reminder of the physical examination were unremarkable.

Hematology and biochemistry were unremarkable and FIV and FeLV serology were negative on the IDEXX^®^ snap FIV/FELV combo test.

CT of the head and neck revealed bilateral increased soft tissue attenuating thickening of the turbinate in both nasal cavities, distributed mostly ventrally and slightly more pronounced in the right nasal cavity. The soft tissue attenuating lesions were not contrast-enhancing. There was no evidence of ethmoidal turbinate loss, and the air passage was not obstructed. The radiologist concluded bilateral non-destructive rhinitis, more commonly associated with non-neoplastic chronic rhinitis, but lymphoma could not be ruled out.

Histopathology of biopsies revealed a locally extensive, marked lymphoplasmacytic infiltrate. There was no evidence of infectious agent or foreign body material. The infiltrate was mixed and mainly composed of small and intermediate-sized lymphocytes (nucleus size ranges from 1.5 to 2 red blood cells with hyperchromatic chromatin and inconspicuous nucleolus), plasma cells and no mitotic figures. Due to the mixed, mainly lymphoid infiltrate, lymphoplasmacytic inflammation was the top differential, but lymphoma could not be ruled out (Figure 4 and Figure 5). Immunohistochemical stains revealed a dense aggregate of CD20 (Thermo Fisher Scientific, Invitrogen, Waltham, MA, USA, PA5-16701) positive-stained B cells (Figure 6), as well as scattered small numbers of individual positive CD3 (Dako, Carpinteria, CA, USA, A0452) T cells. The pathologist concluded a diffuse intermediate-sized B cell lymphoma. A second pathologist reviewed the histopathology slides and confirmed the diagnosis, classifying it as low grade based on the low MI of 0 per ×400 field [15,16,17].

The patient was started on 2 mg chlorambucil three times a week (approximately 6 mg/m^2^ EOD), as well as prednisolone 5 mg daily, progressively tapered down. Rechecks were carried out monthly and no side effects or hematological and biochemistry abnormalities relating to chlorambucil were noticed. The cat markedly improved after starting the treatment and remained in remission for 9 months, until he developed two episodes of suspected full and partial seizures. However, no clinical signs of nasal lymphoma were present. A restaging with blood work, thoracic radiography and CT scan of the head and neck were repeated. CT scan did not reveal any abnormalities in the nasal cavity and cribriform plate was intact. However, a small intracranial extra axial mass (1.5 × 1.0 × 0.4 cm diameter) was found in the left olfactory region of the frontal lobe. The most likely differential diagnoses included an intracranial extension of the lymphoma or a meningioma. The patient was started on levetiracetam and switched from chlorambucil to lomustine and cytarabine, due to better blood–brain barrier penetration. However, during the following two weeks, his neurological signs deteriorated, with more seizure episodes, and the owner elected for euthanasia. Unfortunately, postmortem examination was not permitted, so it was not possible to confirm the nature of the mass in the brain. Chlorambucil and prednisolone-specific DFI and survival time were, respectively, 9 and 10 months.

#### 2.1.3. Case N.3

Aged 16 years and 5 months, a male neutered domestic short-haired cat presented with chronic unilateral (right) mucopurulent nasal discharge of three-month duration and occasional sneezing.

On initial physical examination, he was bright, alert and responsive. The cat was overweight at 6.0 kg, with a BCS of 8/9. The patient had mucopurulent nasal discharge and reduced air flow from the right nostril, but no exophthalmos or facial deformity were noted. Retropulsion of both eyes was negative for mass effects, regional and peripheral lymph nodes were within normal limits and abdominal palpation was unremarkable. On thoracic auscultation, a Grade 2/6 dynamic systolic murmur was present; otherwise, the reminder of the examination was unremarkable.

Case N.3 was found to be in early IRIS Stage 3 non-hypertensive, non-proteinuric chronic kidney disease. FeLV/FIV serology were negative on the IDEXX^®^ snap FIV/FELV combo test. Abdominal ultrasound showed bilateral small kidneys with degenerative changes compatible with chronic kidney disease.

CT of the head and neck revealed rhinitis and sinusitis, with a mild destructive component. There was also a consolidated soft tissue component of the right maxillary recess and a contrast-enhancing nasopharyngeal mass (1.3 × 0.7 × 0.3 cm) with diffuse thickening of the pharynx. CT also revealed bilateral otitis media as a likely extension from the pharyngeal mass. Posterior rhinoscopy revealed numerous pale multinodular polypoid masses throughout the nasopharynx. A nasal flush with saline was performed, but it harvested only moderately thick mucus. Biopsy forceps via rhinoscopy were used to collect samples of the nasopharynx mass.

The histopathology analysis revealed an unencapsulated, poorly demarcated, infiltrative, densely cellular neoplasm. Neoplastic round cells had variable indistinct cell margins. They had a round to oval nucleus of 1.5 to 2 red blood cells in size, finely stippled to finely clumped to hyperchromatic chromatin, and 0 to 2 distinct, small to medium, magenta nucleoli. There were 5 mitotic figures in 10 per ×400 fields. Infiltrating with the neoplastic cells were large numbers of mature lymphocytes and a few neutrophils (Figure 7 and Figure 8). Immunohistochemical stains of nasal mucosal sections revealed all neoplastic cells showing positive immunoreactivity to CD20 (Thermo Fisher Scientific, Invitrogen, Waltham, MA, USA PA5-16701), a marker for B cells (Figure 9), and lack of immunoreactivity to CD3 (Dako, Carpinteria, CA, USA, A0452), a marker for T cells. The histopathology confirmed a diffuse intermediate-sized B cell lymphoma. A second pathologist reviewed the histopathology slides and confirmed the diagnosis, classifying it as low grade with a mitotic index of 0 to 1 per ×400 field [15,16,17].

The cat was started on 2 mg chlorambucil three times a week (approximately 6 mg/m^2^ EOD) and an anti-inflammatory dose of 5 mg prednisolone daily. Clinical signs markedly improved and the prednisolone was eventually reduced and stopped 5 months into chemotherapy. Rechecks were carried out monthly and no side effects or hematological and biochemistry abnormalities related to chlorambucil were reported. At the time of writing, the patient is currently alive and well on chlorambucil, with no obvious recurrence of the lymphoma and no clinical signs, 27 months after starting the treatment.

## 3. Discussion

Examination of the data reported here contributes to our understanding of whether nasal lymphoma with a low mitotic index might respond to an oral chemotherapy protocol using chlorambucil and prednisolone.

In this case series, two of the cats had an intermediate B cell lymphoma and one had a large-cell B cell lymphoma, all with low MI (less than 5 per ×400 field). Despite the classification of intermediate- to large-cell lymphoma, the MI index was relatively low, and based on the classification by Valli et al., they were all considered low grade [15]. All the cases initially also lacked an aggressive clinical presentation, and based on CT scan appearance, they were also not at advanced stages of disease. For all these reasons, the owners were offered the options of standard COP or CHOP protocols or oral chlorambucil. The final decision was made by the owners, also considering their finances and time schedules.

In Case N.1, the owner declined any possibility of weekly rechecks, and the CHOP protocol was only started when no other oral chemotherapy treatment options were available.

In Case N.2, there were no signs of recurrence of the lymphoma in the nasal cavity, but a mass in the brain was later found. From the CT scan appearance and strong contrast uptake, the main differentials were a recurrence of the lymphoma or a primary brain tumor, such as a meningioma. It is quite unusual for nasal lymphoma to recur in the brain without involvement of the nasal cavity and cribriform plate destruction/infiltration, but this possibility could not be ruled out completely. Unfortunately, an MRI of the brain, biopsy or postmortem could not be performed to further confirm the nature of the mass.

All cases reported here had responded to this treatment. Cases N.2 and N.3 (modified Adams Stage 1), achieved a relatively long disease-free interval of 9 and 27 months, respectively [19].

In Case N.1 (classified as Modified Adams Stage 3), the treatment with chlorambucil achieved only a short response of 3 months. The CHOP protocol was started only as a third-line treatment, and it is possible that starting a more aggressive CHOP chemotherapy protocol as a first-line treatment could have achieved a better outcome by delaying the occurrence of multidrug chemotherapy resistance. However, CHOP protocol treatment achieved only a short disease-free interval (DFI), similar to the initial chlorambucil and later lomustine-based protocols, so it is unlikely that the CHOP protocol as a first-line treatment would have achieved a markedly longer DFI. The cause of pancytopenia could not be ascertained as a bone marrow biopsy or postmortem examination were not permitted.

Despite being defined as low grade based on the classification by Valli et al. [15,16,17], Case N.1 had a more rapid progression and more aggressive clinical behavior, which is in line with a previous study that concluded that feline nasal lymphomas are clinically aggressive, with no significant correlation of mitotic index with mortality [7]. It is interesting to note that, in this case, despite the MI being considered low by Valli et al.’s classification [15,16,17] (3–4 per ×400 field), it was still much higher compared to Cases N.2 and N.3, respectively (0 per ×400 field and 0–1 per ×400 field). This means that Case N.1 had still at least a tenfold higher MI per ×400 field compared to Cases N.2 and N.3. It is possible that previously published criteria on canine lymphoma cannot be applied to grade feline nasal lymphomas, and different MI cut-offs may be taken into consideration for their categorization [15,16,17].

The response to chlorambucil treatment in our case series (for Cases N.2 and 3), in combination with one case report previously published [14], could be suggestive of the existence of a subset of indolent nasal lymphomas that could achieve a long response to chlorambucil and prednisolone. The clues regarding the indolent nature of a nasal lymphoma could lie in a significantly low MI (0–1 per ×400 field) in combination with the lack of an aggressive clinical presentation and appearance on CT scan.

It cannot be completely ruled out that Case N.2 was in fact lymphoplasmacytic rhinitis misdiagnosed as low-grade lymphoma. However, all the histopathology slides were reviewed by two pathologists and the diagnosis confirmed with immunohistochemistry. The presence of lymphoplasmacytic rhinitis in combination with the lymphoma was considered the main reason for doubts in the first histopathology evaluation. The most common feline low-grade lymphoma affects the intestine, and low-grade lymphomas are rarely reported for other anatomical forms [4]. Low-grade intestinal lymphomas are of T cell origin with an epitheliotropic growth pattern [20]. Low-grade nasal lymphoma is rarely reported in cats [4]. The cases that we reported here, both the intermediate- and large-cell lymphomas, were confirmed by histopathology and immunohistochemistry to be all of B cell origin. None of the cases were of T cell origin or showed the typical epitheliotropic growth seen in low-grade intestinal, mucosal, cutaneous or nasal lymphoma [7,20].

In Cases N.2 and 3, the lack of advanced stage of the nasal tumor, and the lack of severe clinical signs at presentation, were other characteristics that indicated the possibility of offering a more conservative and less intensive treatment approach. It is possible that the long DFI, particularly in Case N.3, could be due to the early stage of the disease, and it is possible that, at earlier stages, some cases could be treated more conservatively with a reasonably good outcome. In Case N.1, the decision of starting the cat on chlorambucil and prednisolone was dictated by the relatively low MI according to the Valli classification and the owner’s wishes. The treatment in this case was not successful and it is possible that an early, aggressive treatment with multi-agent chemotherapy could have resulted in a better outcome, as cancer cells with a multidrug resistance phenotype could have been selected after chlorambucil exposure. It is also possible that long-term treatment with prednisolone could have caused the development of cancer cell chemotherapy resistance. However, cats pre-treated with prednisolone do not seem to develop obvious chemotherapy treatment resistance [8,21,22].

In conclusion, two out of three cases reported here suggest chlorambucil and prednisolone as a first-line treatment in selected cases with a more indolent clinical course, such as early-stage nasal lymphoma associated with a very low mitotic activity (0–1 ×400 field).

## Figures and Tables

**Figure 1 vetsci-09-00472-f001:**
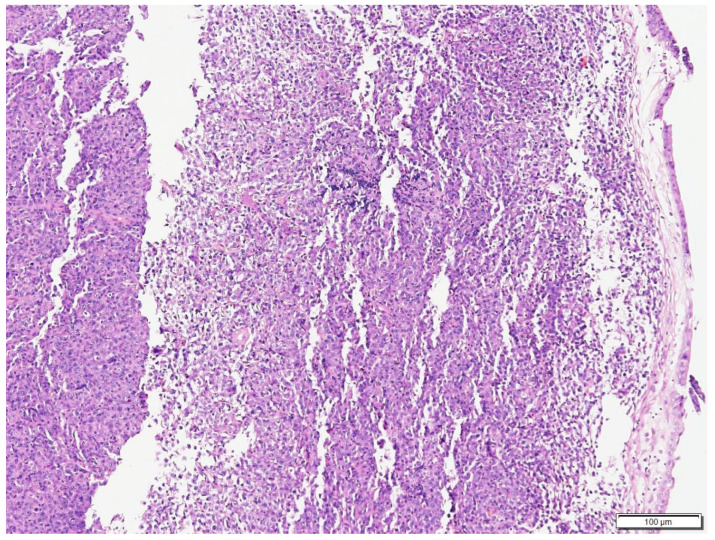
Histologic findings of the mass at the choanae of Case N.1. The submucosa is expanded by a neoplasm composed of solid sheets of round cells. Hematoxylin and eosin (H&E) stain (100× magnification).

**Figure 2 vetsci-09-00472-f002:**
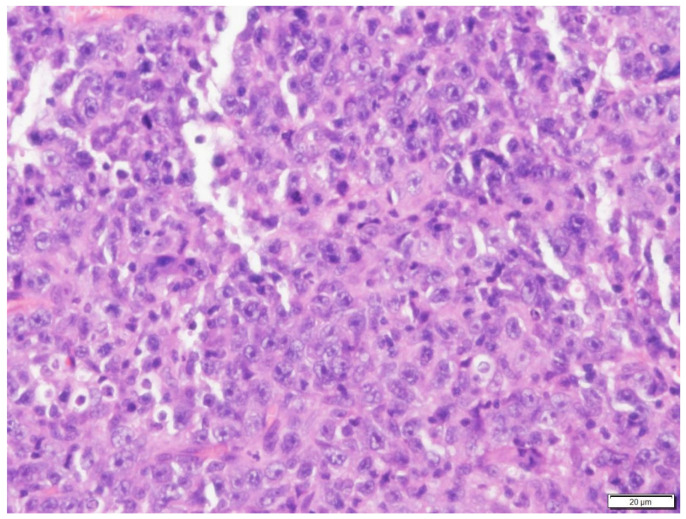
Higher magnification of the neoplastic cells in Case N.1. The cells were round to elongated, with minimal cytoplasm and vesicular pleomorphic nuclei and 1 to 2 distinct, magenta, medium nucleoli. H&E stain (400× magnification).

**Figure 3 vetsci-09-00472-f003:**
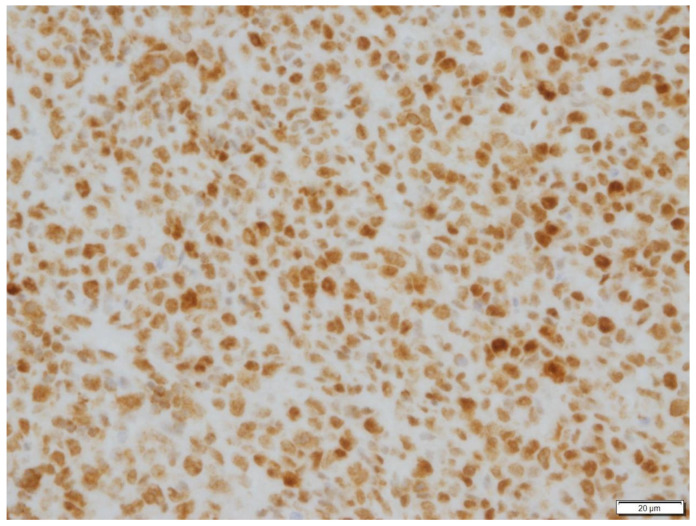
Immunohistochemical stain of neoplastic cells in Case N.1. Neoplastic cells exhibited strong immunoreactivity for CD79a. Immunohistochemical stain CD79a (400× magnification).

**Figure 4 vetsci-09-00472-f004:**
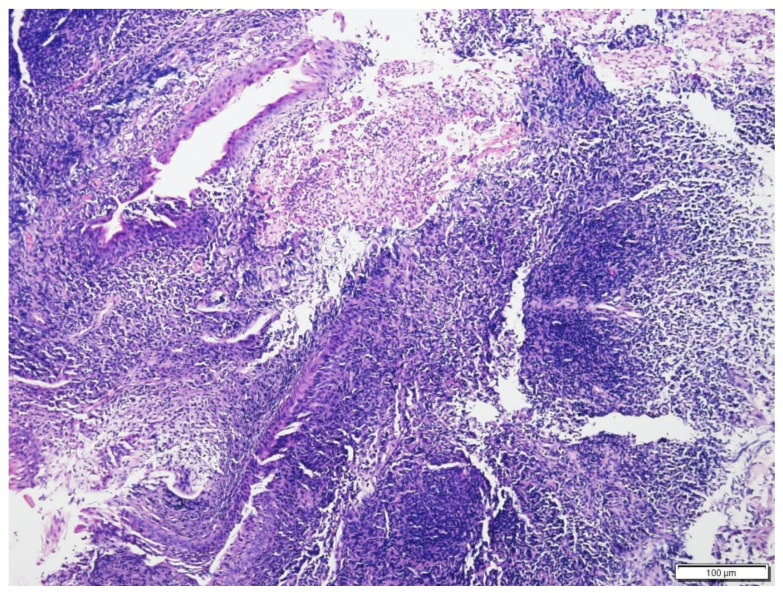
Histologic findings of nasal biopsy of Case N.2. The submucosa is expanded by a dense infiltrate of mixed small and large lymphocytes and plasma cells, and scattered with areas of necrosis. H&E stain (100× magnification).

**Figure 5 vetsci-09-00472-f005:**
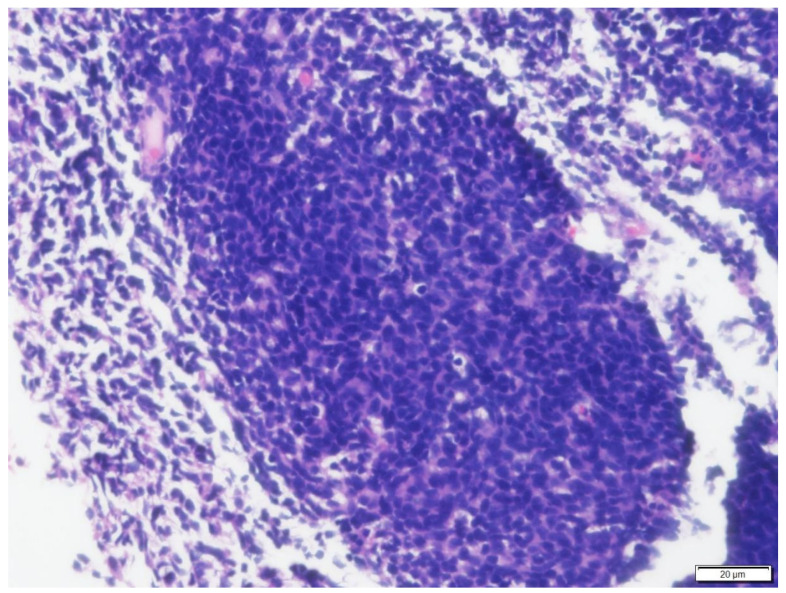
Higher magnification of the dense infiltrate in Case N.2. The large lymphocytes were round, with minimal cytoplasm, a round to oval nucleus, hyperchromatic chromatin and inconspicuous nucleolus. H&E stain (400× magnification).

**Figure 6 vetsci-09-00472-f006:**
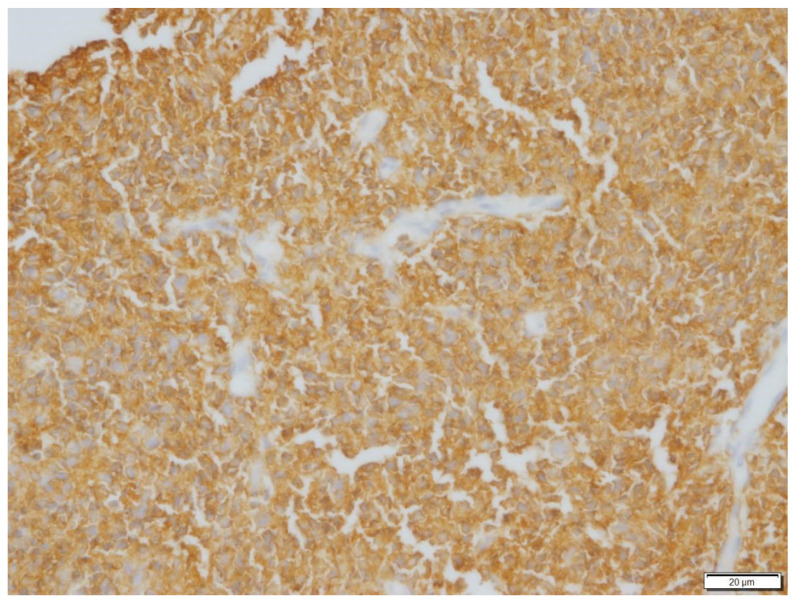
Immunohistochemical stain of neoplastic cells in Case N.2. Neoplastic cells exhibited strong immunoreactivity for CD20. Immunohistochemical stain CD20 (400× magnification).

**Figure 7 vetsci-09-00472-f007:**
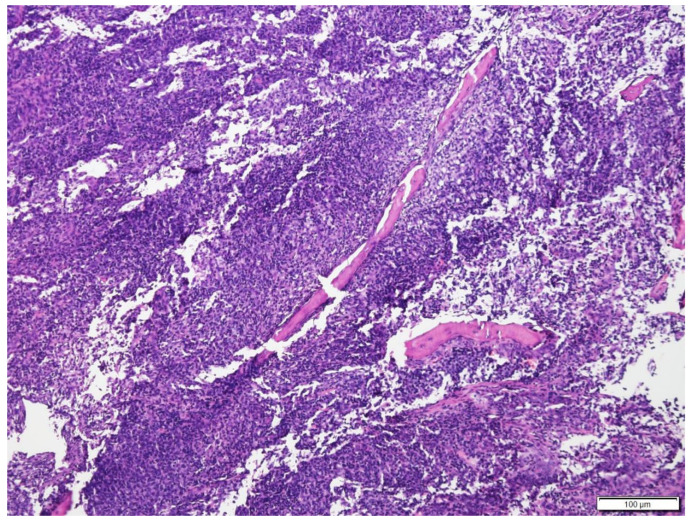
Histologic findings of nasal biopsy of Case N.3. The submucosa is expanded by dense sheets of neoplastic cells separating the turbinate bone spicules. H&E stain (100× magnification).

**Figure 8 vetsci-09-00472-f008:**
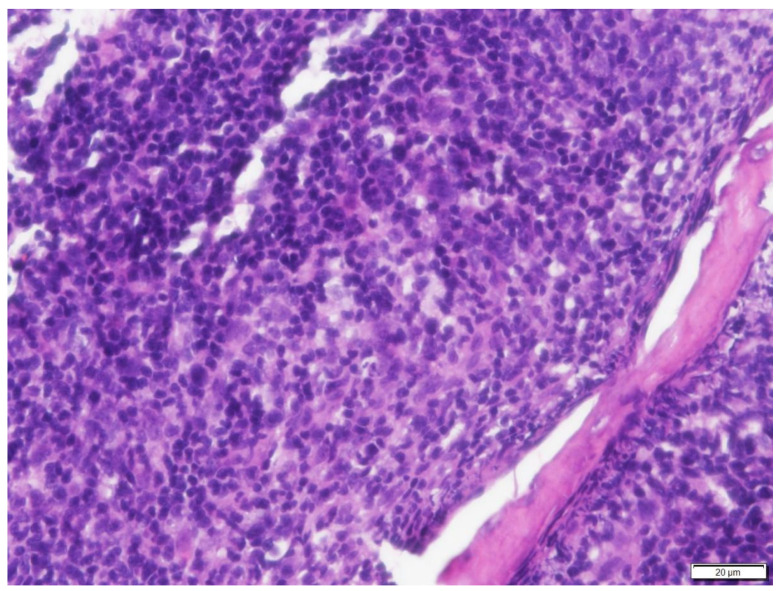
Higher magnification of the neoplastic cells in Case N.3. Neoplastic round cells had round to oval nucleus, finely stippled to finely clumped to hyperchromatic chromatin and 0 to 2 distinct, small to medium, magenta nucleoli. Infiltrating among neoplastic cells were variable numbers of small mature lymphocytes and neutrophils. H&E stain (400× magnification).

**Figure 9 vetsci-09-00472-f009:**
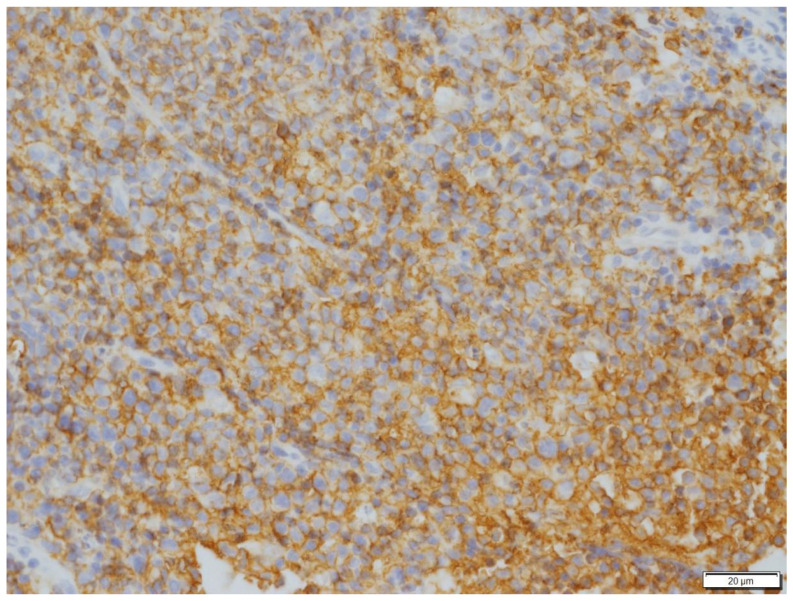
Immunohistochemical stain of neoplastic cells in Case N.3. Neoplastic cells exhibited strong immunoreactivity for CD20. Immunohistochemical stain CD20 (400× magnification).

## Data Availability

Not applicable.

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
