# Peer review of "Nasal Lymphoma with Low Mitotic Index in Three Cats Treated with Chlorambucil and Prednisolone"

_vetsci, 2022, doi:10.3390/vetsci9090472_

Round 1
Reviewer 1 Report (Previous Reviewer 1)
Dear Authors,
You have improved the quality of the manuscript and accepted the suggested changes. I have only some minor suggestions.
I wish you all the best.
---
Abstract:
“This case report series showed early but weak evidence”. Confusing phrase.
Introduction
Lines 75-80: I still think that this part belongs to the Material and Method Section. Also, lines 80-82 belong to the Results section.
I suggest including the company name of the immunomarkers used in the Material and Method section.
I suggest replacing Figure 6 since the histopathologic quality is doubtful.
Author Response
Please see the attachment. Thank you.

Reviewer 2 Report (Previous Reviewer 3)
The reviewer would thank the authors for their efforts in addressing all the reviewer’s concerns. The manuscript is improved. However, a few changes are needed before publication in the reviewer’s opinion.
The inclusion of cat n.1 seems not in line with what the authors try to say with this manuscript. In comparison with cat n.2 and 3, cat n.1 presented bad clinical condition, higher MI (ten fold), large cells, FIV infection and potential nasopharyngeal involvement. Of the 39 cases of Santagostino’s paper only 15% had MI>3/HPF and 22/39 had MI<1/HPF. The authors of that paper were not able to find any association between MI and survival, however cats receive no treatment, making any further discussion about responsiveness to therapy and survival rate not possible. I think case 2 and 3 are similar in presentation, imaging and histopathological findings and outcome making stronger the conclusion the authors try to say that chlorambucil would be an option in feline nasal lymphoma associated with low mitotic index. If the authors would decide to not delete case n.1 from this case series, I would suggest them to further discuss why they think case n.1 did not benefit from chlorambucil and they should convince the reader to think about chlorambucil when he/she will have to face with feline nasal lymphoma associated with MI>3.
Title: I would suggest the authors to not to use the word “low grade” especially if they include case n.1. I will use low mitotic index.
Line 42-43 Reading lines 43-49 the reader understand that survival time ranges from at least 365 to 750 days. However lines 42-43 stated 116-358 days citing ref 4,9 and 10. It is confusing to me. I would suggest to rewrite the sentence (for example “Survival time reported with COP or CHOP chemotherapy depends on the study and it is reported up to 750 days 4,9,10”)
Line 81 please delete “cell” before “B-cell”
Line 95-95 Please state which drug the cat was taking for her hyperthyroidism
Line 100 Could the authors add information on mass size? I would add the three max diameter
Figure 1 and 2. Please add magnification
Case n.1 If I read well the cat had a mass involving the nasopharynx making it a possible Adam’s stage III. Please verify with a radiologist if an Adam’s stage I is appropriate for this case. Unfortunately Adam’s staging lack of categorization of nasal tumor with nasopharyngeal involvement when it is minimal. However, nasopharyngeal mass are potentially more aggressive/less responsive. An Adam’s stage III could also worsen life expectancy for cat 1. However, as I suggested above, I would remove case n.1 from this case series.
Case n.3 was the cat still under chlorambucil at the time of writing (27 mo)? Please specify
Discussion section: I would move 310-319 before line 301 “In case N.2, there were…..”
Lines 329-331 The sentence “It is possible that in feline nasal lymphoma, a much lower MI cut-off may take into consideration for categorizing low-grade lymphoma, compared to the previous published classification” is not completely correct. The previous published classifications the authors referred to are for canine lymphoma. I would reword the sentence like “It is possible that previous published criteria on canine lymphoma would not be applied to grade feline nasal lymphomas and different MI cut-offs may take into consideration for their categorization”
BCS of 2/9 of case n.1 associated with FIV infection may explain the short survival time. Cachexia is an independent negative prognostic factor in people even if not associated with a tumor.
Conclusions are not supported by authors’ findings.
“In conclusion, chlorambucil and prednisolone shows potential to be considered as first-line treatment in selected cases”, which cases? With MI<5/HPF? Seeing Santagostino’s paper almost all feline nasal lymphomas have MI<5/HPF. With large or intermediate cells? In this case series large cell lymphoma seemed to not respond to treatment.
“… with early-stage nasal lymphoma..” See above regarding case n.1 and nasopharyngeal involvement
“…with low mitotic activity…” again seeing Santagostino’s paper almost all feline nasal lymphomas have MI<5/HPF.
Author Response
Please see the attachment. Thank you so much for your help in reviewing our paper.

Round 2
Reviewer 2 Report (Previous Reviewer 3)
Dear Authors, thank you for your job and your efforts. As I can read now, your study findings support conclusion and title well introduces your study.
I have a few minor editing to suggest:
line 491 to be consistent please change "whether low grade nasal lymphoma" with "whether nasal lymphoma with low mitotic index"
line 615 I would rewrite the sentence like this "In conclusion, two out of the three cases reported here suggest chlorambucil and prednisolone as first-line treatment in selected cases with a more indolent clinical course, early-stage nasal lymphoma associated with a very low mitotic activity (0-1x 400 field)."
Author Response
Dear Reviewer 2,
We have changed the sentences as you have suggested.
There was no change to the first suggestion.
The second suggestion, I have added the words "such as" into the sentence but have kept the rest of the sentence the same.
Thank you once again for your effort and help in reviewing our article. It is much appreciated.
Sincerely,
Karen
This manuscript is a resubmission of an earlier submission. The following is a list of the peer review reports and author responses from that submission.
Round 1
Reviewer 1 Report
Review Report
Nasal lymphoma in 3 cats treated with chlorambucil and prednisolone.
No: VetSci 1764557-
Journal: Veterinary Sciences
Article Type: Case Series
The current study addresses an important topic and is of general interest to the readers of the journal: describe the use of chlorambucil and prednisolone in three feline intermediate to large B-cell, low-grade nasal lymphomas. The importance of this study relies on the fact that (1) it is unknown if low-grade nasal lymphoma should be treated differently from inter-mediate or high grade and (2) chlorambucil and prednisolone is effective in the long-term management of feline low-grade intestinal lymphoma.
I found the paper interesting and well-written. I am convinced that the authors had made a great effort in this study. I congratulate the authors and my suggestions are only in some minor aspects.
I took the liberty to review the document directly since it lacks line numbers, which makes the suggestions difficult to describe.
----
Suggestions:
Title: I suggest including “Low grade” Nasal lymphoma in 3 cats treated with chlorambucil and prednisolone.
Introduction:
- Figures: I suggest moving the figures accordingly to cases descriptions. Also, I suggest including figures of the immunohistochemistry, if available.
-Conclusions: I suggest reformulating the conclusion since it is clear that a larger study needs to be performed before stating this protocol as a first line. The study showed early but weak evidence (2 in 3 cases) that the protocol may have potential.

Reviewer 2 Report
Dear authors, congratulations for the work done, unfortunately, however, I note some important critical issues that I set out below:
- the casuistry is really poor
- there is not even a minimal control group
- the sample is not homogeneous and even case n3 has a concomitant pathology
- precisely in case n3 with iris staging 3 he underwent therapy with prednisolone 5mg / day why?
- could the results obtained be related to prednisolone alone and not to chlorambucil?
- in case n2 you write that you have climbed the therapy but do not explain how.
Best regards
Reviewer 3 Report
Reviewer comment:
The present study reports the use of chlorambucil and prednisolone in cats diagnosed with feline lymphoma.
Biases in inclusion criteria and other major flews make me to unfortunately reject the study. However, since no study have reported the efficacy of chlorambucil-prednisolone in cats with nasal lymphoma, the study would be reevaluate after extensive revision
The authors wanted to include only low grade lymphoma based on a low MI of less than 5 mitoses in 10 HPF however, case n.1 seemed misincluded.
Introduction:
“Survival time reported with COP or CHOP chemotherapy depends on the study and it is reported between 116–358 days [4,8,9]. Intensive chemotherapy treatment carries some risks of side effects, around 83% of patients develop mild side effects [10] and a small …” In my opinion, the authors should better define study findings they cited. Taylor et al reported >70% of complete response for cats treated with chemo with approximatively 750 days of MST in these cases. Overall MST was influenced by partial or not-responded cats. Teske et al reported >70% of CR with 75% 1-year survival expectancy. MST of 536 days was reported by Haney et al.
The sentence “The low-grade classifica- tion was based on that of Valli et al. [14] which considers low grade lymphoma as tumors with a low MI of less than 5/10HPF despite intermediate to large cell lymphoma being usually classified as high grade.” should be moved in M&M section as it include method of inclusion. The reviewer was not able to find where in ref [14] Valli et al. suggested <5/10HPF MI as cut-off for low grade lymphoma.
The sentence “Two of the cats had an intermediate cell size lymphoma and one large cell lymphoma, but all with low MI (less than 4-5/10 HPF)” should be moved in the case description
Please state which test was performed to say “She was feline leukaemia virus (FeLV) negative, but feline immunodeficiency virus (FIV) positive”
Please add weight of cats
Case N1. “The mitotic rate was 3-4 per single high-powered field”, this mitotic rate is huge, it doesn’t look low grade in my opinion. Please justify your interpretation
Please provide reference and protocol schedule of LPP protocol. Did the authors intend LPP instead of LLP as they wrote?
What did the authors base their evaluation regarding progressive disease on?
I would remove the authors’ etiological hypotheses regarding pancytopenia in this cat from this section.
Case N2. “nucleus size ranges from 1.5 to 2 red blood cells” sound intermediate cells to me and not large.
Case N3 can the authors substage IRIS CKD staging of this cat (proteinuria and blood pressure)?
Why did the cat stop chlorambucil after 5 months?
Please stage nasal disease with the Adam’s staging system or subsequent staging systems. Please mention the reference you will use.
Discussion
The authors still consider MI 3-4/HPF low. However it is quite high, this tumor shouldn’t be associate with a very low malignancy profile like the other 2. The sentence “The low-grade classifica- tion was based on that of Valli et al. [14] which considers low grade lymphoma as tumors with a low MI of less than 5/10HPF despite intermediate to large cell lymphoma being usually classified as high grade” the authors mentioned in the introduction should mean <5 total mitoses in 10 HPF that should be equal to 0-1 mitosis x HPF.